# Analog–digital hybrid computing with SnS$_2$ memtransistor for low-powered sensor fusion

Shania Rehman[1], Muhammad Farooq Khan[1], Hee-Dong Kim[1] & Sungho Kim [1✉]

Algorithms for intelligent drone flights based on sensor fusion are usually implemented using conventional digital computing platforms. However, alternative energy-efficient computing platforms are required for robust flight control in a variety of environments to reduce the burden on both the battery and computing power. In this study, we demonstrated an analog–digital hybrid computing platform based on SnS$_2$ memtransistors for low-power sensor fusion in drones. The analog Kalman filter circuit with memtransistors facilitates noise removal to accurately estimate the rotation of the drone by combining sensing data from the gyroscope and accelerometer. We experimentally verified that the power consumption of our hybrid computing-based Kalman filter is only 1/4$^{th}$ of that of the traditional software-based Kalman filter.

---

[1] Department of Electrical Engineering and Convergence Engineering for Intelligent Drone, Sejong University, Seoul 05006, Korea. ✉email: sungho85kim@sejong.ac.kr

Sensor fusion is a widely used technique in various control systems that aims to overcome the limitations of individual sensors by gathering and fusing data from multiple sensors to produce more reliable information with less uncertainty[1]. Sensor fusion has been actively applied in unmanned aerial vehicles (UAVs), also referred to as drones. To determine the accurate position or orientation of the drone in real time, various types of sensors, such as global positioning systems, gyroscopes, accelerometers, magnetometers, and pressure sensors, are embedded in the drone. Because these sensors are prone to errors, including noise and drift, sensor fusion is essential for achieving an optimal accuracy from noisy sensing data[2].

For robust flight, sensor fusion to estimate the Euler angle of a drone has been commonly exploited in all drones. Euler angles provide a method to represent the three-dimensional orientation using a combination of three rotations about different axes. In particular, the rotations of the drone are referred to as roll ($\phi$), pitch ($\theta$), and yaw ($\psi$) (Fig. 1a). A gyroscope and accelerometer are used to measure the Euler angles of the drone (Fig. 1b and Supplementary Note 1)[3]. The gyroscope measures the rate of rotation projected on its sensing axis, that is, the angular velocities ($p$, $q$, and $r$). By substituting the measured angular velocities into Eq. (S1) and then performing integration, the Euler angles of the drone can be obtained. However, the gyroscope exhibits a steadily growing error over time because noise accumulates during the integration[3]. Therefore, the application of a gyroscope alone cannot provide an absolute measurement of rotations. Alternatively, the accelerometer can measure linear accelerations along three axes ($A_x$, $A_y$, and $A_z$) and provide Euler angles using Eq. (S5). However, the accelerometer is likely to be subject to high levels of noise owing to vibrational effects from the motor of the drone[3].

The Kalman filter[4], which is a recursive algorithm for providing the best estimate (filtered output) from noisy sensing data (raw input), is widely used for sensor fusion[5]. Variants of the Kalman filter algorithm have shown excellent performance in noise reduction, even in nonlinear systems[6]. The algorithm involving complex matrix operations should be executed more than tens of times per second; however, the computing power and available memory of the microcontroller embedded in the drone are limited because of insufficient battery capacity. Moreover, as drones are evolving to be applied for more complex missions, a more complex sensor fusion algorithm is required. The growing demand for computing power with limited battery capacity restricts higher degrees of freedom in drone operation. Therefore, the development of an alternative computing platform with a higher energy efficiency for sensor fusion is essential for future drone technology[7].

Notably, an analog-digital hybrid computing platform, which is inspired by biological neural networks (typically referred to as neuromorphic systems[8]), has been considered as a promising candidate for realizing energy-efficient computing[9–13]. The precisely tunable analog resistive switch (i.e., memristor) energy-efficient analog computation with a process-in-memory architecture and also allows for functional reconfigurability. The feasibility of the analog-digital hybrid computing platform has been successfully demonstrated to mitigate the computational burden of vector-matrix multiplication in the calculation of various machine-learning algorithms[14–16]. The research to reduce overall energy consumption by replacing a part of digital calculation with analog circuits is being conducted in various application fields. Furthermore, recent advancements in memristors based on two-dimensional materials offer the possibility of designing new materials with atomic-level precision, resulting in excellent resistive switching performance with only a small amount of energy consumption[17–20]. A drone is a complex real-time sensing system that can benefit substantially from this memristor-based analog-digital hybrid computing platform. Because the Kalman filter algorithm can be expressed by linear

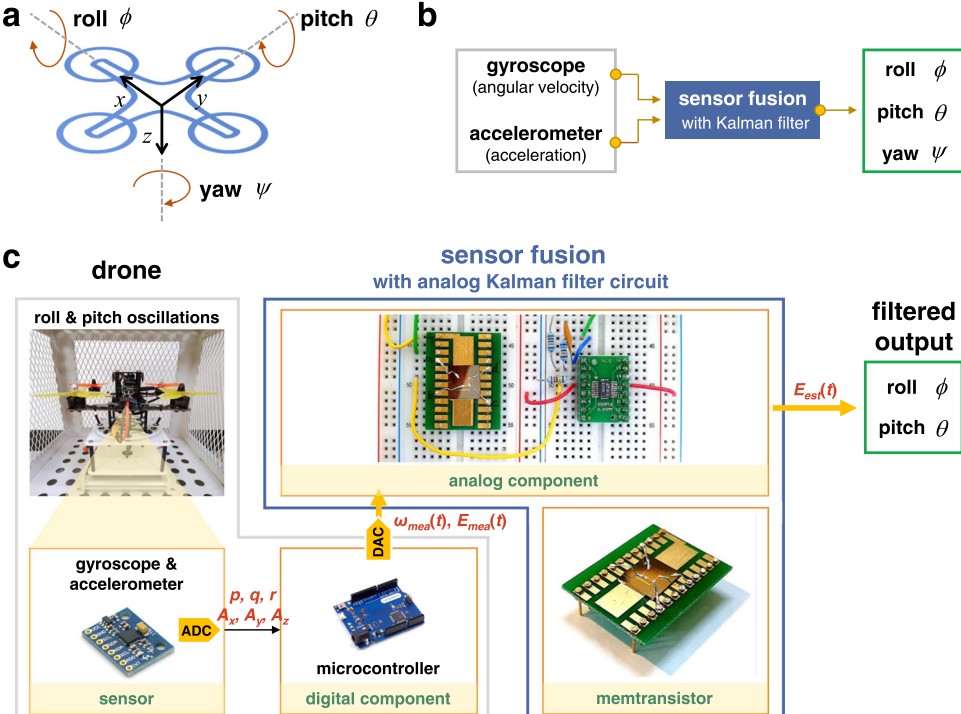

**Fig. 1 Overview of memtransistor-based analog–digital hybrid computing platform. a** The inertial frame of the drone. **b** The Euler angle estimation procedure through the sensor fusion with Kalman filter algorithm. **c** Schematic illustration of our analog–digital hybrid computing platform. The analog component, i.e., memristor-based analog circuit, is responsible for executing the Kalman filtering algorithm. The digital component, i.e., microcontroller, contributes to the data collection from the sensor module, and signal generation through the DAC.

equations, it can be implemented using memristor-based analog circuits. Moreover, this memristor-based analog component can operate independently without using computing resources from the digital processor, thereby reducing the computational load of the digital component. Nevertheless, there exist some demonstrations of memristor-based hybrid computing for drones but only one recent study has applied this to control an inverted pendulum for a mobile robot[21].

In this study, we demonstrate a memtransistor (memristor with transistor structure[22])-based analog-digital hybrid computing platform for sensor fusion with higher energy efficiency. The measured data from both the gyroscope and accelerometer were combined to accurately determine the Euler angles of drones, wherein the Kalman filter algorithm was implemented using a customized analog circuit with the memtransistor. Because this analog Kalman filter circuit can operate independently without using the computing resources of the microcontroller, the computational burden on the microcontroller is reduced, and subsequently a reduction in overall power consumption can be expected. Here, we used transition-metal dichalcogenide (TMD) materials in the channel of the memtransistor, which is a three-terminal hybrid memristor and transistor. The bulk traps located at the tin disulfide ($SnS_2$) nanosheet exhibit a highly reliable nonvolatile resistive switching behavior, which is achievable through the electrical pulse applied to a gate electrode. The precise tunability of the $SnS_2$ memtransistor allows for the reconfigurability of our analog-digital hybrid computing platform. Finally, we experimentally demonstrated that a drone using our hybrid computing performs sensor fusion with higher energy efficiency than a drone with only a conventional digital processor.

## Results

**Experimental conditions**. Figures 1c and S1 show our home-built quadrotor, which is composed of a microcontroller (ATmega32U4) and sensor module (MPU6050, including a gyroscope and accelerometer). Detailed specifications of the microcontroller and sensor module are presented in Supplementary Note 2. In our experiment, sinusoidal oscillations of ±30° were performed about the roll and pitch axes simultaneously with a frequency of 0.2 Hz (for simplicity, the yaw angle was fixed to zero). Thereafter, the angular velocities ($p$, $q$, and $r$) and accelerations ($A_x$, $A_y$, and $A_z$) were measured by the gyroscope and accelerometer, respectively, with a sampling rate of 100 Hz (the measured raw data are shown in Figs. S2a, S2b respectively). Because the gyroscope is subject to bias instabilities, its initial zero reading will cause a drift over time owing to the integration of inherent imperfections (Fig. S2c). Similarly, the vibration owing to the high-speed motors resulted in substantial high-frequency noise in the measured accelerations (Fig. S2d). Therefore, neither the gyroscope nor accelerometer can be used alone to accurately estimate the Euler angles; thus, sensor fusion is essential.

In conventional drones, a discrete-time Kalman filter algorithm is performed on the microcontroller (i.e., software-based Kalman filtering, whose principles are summarized in Supplementary Note 3). For software-based Kalman filtering, one cycle of the algorithm shown in Fig. S4 should be executed to update the Kalman gain ($K$) whenever new sensing data is generated at each sampling time. Consequently, dozens of algorithm cycles per second should be executed continuously for sensor fusion, which consumes a substantial amount of power, and causes a high latency[23]. Therefore, we demonstrated a memtransistor-based analog Kalman filter circuit, that is, a continuous-time hardware-based Kalman filter (Fig. 1c). It has been proven that the Kalman gain converges to a constant value after several algorithm cycles[24]. Therefore, in our proposed Kalman filter, a fixed Kalman gain value

was employed rather than a value that requires updating for every algorithm cycle. Thereafter, the constant Kalman gain value is stored using the nonvolatile conductance of the memtransistor; therefore, it can be maintained without additional energy consumption. Moreover, the analog Kalman filter circuit (analog component) can operate independently without using the computing resources of the microcontroller (digital component), thereby reducing the computational burden on the microcontroller.

**Resistive switching characteristics of $SnS_2$ memtransistor**. A memtransistor based on layered $SnS_2$ was employed for our hybrid analog–digital computing platform (see Fig. 2a and the Methods section). Figure 2b shows a high-resolution microscopic image of the memtransistor obtained via a transmission electron microscope (TEM), where the thickness of the $SnS_2$ layer is ~20 nm. The energy dispersive X-ray analysis confirms the presence of sulfur (S) and tin (Sn) in the sample, as shown in Fig. S5. Note that the TEM image reveals a clear crystalline lattice of the $Al_2O_3$ layer, meanwhile, the $SnS_2$ layer is composed of poly-crystalline. The intrinsic defects due to the grain boundaries in the $SnS_2$ layer will provide electrical traps (to be discussed in next Section). Figure 2c shows the transfer characteristics (drain current–gate voltage: $I_D−V_G$) of the $SnS_2$ memtransistor (the source electrode is always grounded). Note that pulsed current–voltage ($I−V$) measurements were used to exclude any bias-cumulative effect. As shown in the inset of Fig. 2c, voltage pulses were applied only for a short period (1 ms) to the gate and drain, and the drain current was measured only when a pulse was applied. Because a long interval time (100 ms) between pulses prevents any accumulation effect, the measured $I_D–V_G$ result shows a hysteresis-free curve. From this hysteresis-free transfer characteristic, the field-effect mobility ($\mu$) of $SnS_2$ was calculated using the following equation:

$$\mu = \frac{dI_D}{dV_G} \frac{L}{WC_{Al2O3}V_D} \qquad (1)$$

where $L$ and $W$ are the length and width of the $SnS_2$ channel, respectively, and $C_{Al2O3}$ is the gate-insulator capacitance per unit area (assuming that the dielectric constant of the $Al_2O_3$ layer is 7.0)[25]. The obtained $\mu$ is 2.18 $cm^2V^{-1}s^{-1}$ at room temperature, which is comparable with that of recently reported $SnS_2$ transistors[26]. In addition, the output characteristics (drain current –drain voltage, $I_D–V_D$) at different gate voltages (Fig. S6a, b) confirm the sufficiently low contact resistance between the source/drain electrodes and $SnS_2$ layer.

Interestingly, when the duration of the applied $V_G$ pulse was increased beyond a certain level (>5 ms), the conductance of the $SnS_2$ channel ($G$) could be modulated. Figure 2d shows the $G$ modulation behavior, in which $G$ can be adjusted gradually by repeatedly applying $V_G$ pulses. To update $G$ (i.e., potentiation or depression of $G$), a $V_G$ pulse of −10 or +8 V was applied for 50 ms along with the grounded source and drain. Subsequently, $G$ can be adjusted gradually by applying $V_G$ pulses in the range of 0.1–1.1 µS. Even after $10^5$ updating cycles, stable resistive switching behavior can be maintained, which allows for the reliable operation of our analog-digital hybrid computing platform. In addition, by exploiting the previously developed update–verify feedback method (Fig. S7 in Supplementary Note 4)[27,28], $G$ can be further tuned precisely within the desired target error range (e.g., $G_{var}$ is set to ±5% in Fig. 2e). Furthermore, each conductance state showed a stable retention performance even after $10^5$ s (Fig. 2f). The precise tunability and stability of $G$ are the basis for the reliable reconfigurability of our analog–digital hybrid computing.

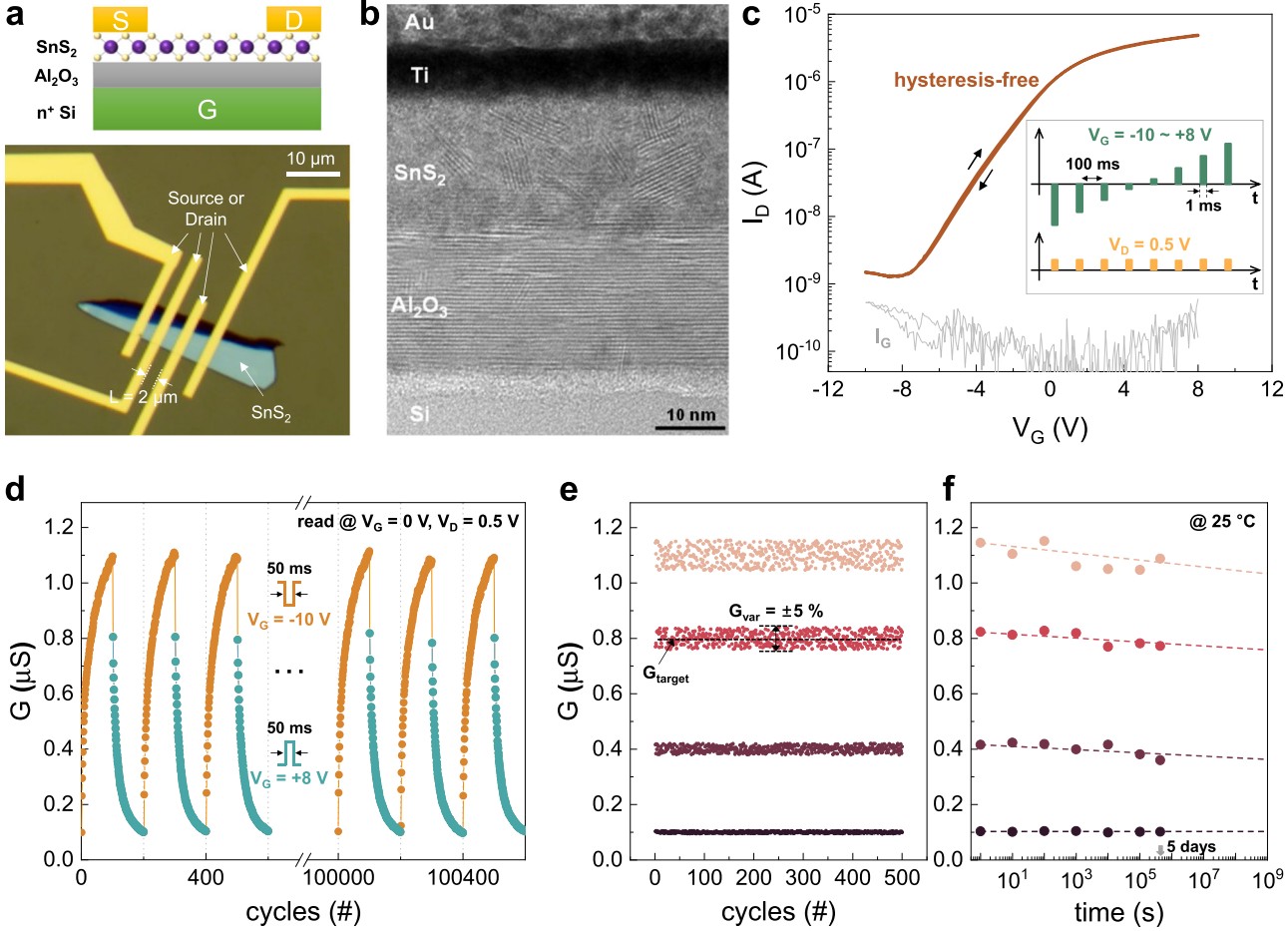

**Fig. 2 SnS₂-based memtransistor. a** Schematic and optical microscopic image of the fabricated SnS₂ memtransistor, where the gate length and channel width are 2 μm and 8 μm, respectively. **b** Transmission microscope image of the device. The thickness of the layered SnS₂ is ~20 nm. **c** Transfer characteristic of the SnS₂ memtransistor obtained through the pulsed I-V method. The inset figure shows the timing condition for the pulsed I-V measurement. Gray curve represents the gate current ($I_G$), which is always below the 1 nA level. **d** Measured conductance modulation behavior of the SnS₂ memtransistor. Each pulse train consists of 100 depression/potentiation pulses ($V_G = -10$ V or $+8$ V for 50 ms) applied to the gate, followed by nonperturbative read voltage pulses ($V_G = 0$ V for 50 ms) within the provided intervals. **e** Four different G states in the SnS₂ memtransistor obtained via the update-verify feedback method when the pre-defined target $G_{var} = 5\%$. **f** Retention property for different G states.

Different resistive switching mechanisms of TMD-based memtransistors have been considered in previous studies[29,30] owing to the following reasons: (1) the adsorption of water and oxygen molecules on the exposed TMD surface[31], or (2) the trapped charges at the TMD/dielectric stack[32]. In this study, we suspect that trapped charges are the origin of such resistive switching in the SnS₂ memtransistor. Our measurement result did not support an adsorbate-mediated mechanism because there was no difference in the switching behavior even when the measurement was performed under vacuum conditions (Fig. S8). In general, various traps have been considered for TMD/dielectric stacks, as follows: interface traps[33] located at the TMD/dielectric interface, border traps[34] located inside the dielectric, or bulk traps[35] located inside the TMD layer. In the case of our SnS₂ memtransistor, because the Al₂O₃ dielectric layer has a crystalline lattice structure as shown in Fig. 2b, the effect from trapped charges in the border traps are negligible. Instead, the interface and bulk traps caused due to the grain boundaries in the SnS₂ layer are expected to be responsible for resistive switching. To investigate which type of trap dominantly contributes to resistive switching, transient I–V measurements were performed to extract the time response of the traps. Figure 3a shows the gate and drain pulses applied to the SnS₂ memtransistor. When $t < 0$, the high negative gate bias ($-8$ V) depleted the

channel. During this period, electrons with energy levels lower than the Fermi level are initially filled in both interface/bulk traps, as shown in Fig. 3b (the estimation of the energy band alignment and Fermi-level position are discussed in Supplementary Note 4 with Fig. S9). When $t \geq 0$, a positive gate pulse ($+8$ V) is applied for a duration of $t_p$. This pulse leads to the carrier trapping and diffusion processes, sequentially:[36] (1) the electrons from the inverted channel are quickly trapped in the interface traps, (2) followed by the diffusion of these trapped electrons toward the bulk trap inside the TMD layer over time. Because two sequential processes have different time constants, the change in $I_D$ also appears over two stages. When $t_p$ is sufficiently short, only the trapping process in the interface trap occurs, which is responsible for the fast $I_D$ transient. Conversely, as $t_p$ becomes longer, the diffusion to the bulk traps also contributes to $I_D$ change, resulting in a slow $I_D$ transient. In other words, traps have a certain time constant and will not respond when the ON pulse width ($t_p$) is shorter than this value[37]. Therefore, extracting the trap time constant with different $t_p$ can reveal which type of trap contributes to the trapped charge-related resistive switching behavior in the SnS₂ memtransistor.

Figure 3c shows the measured $I_D$ transients for different values of $t_p$. When $t_p$ is shorter than 1 ms, the $I_D$ curve can be fitted with a single trap time constant (red line), yielding $I_D = I_0 + A \cdot \exp(-t/\tau_i)$,

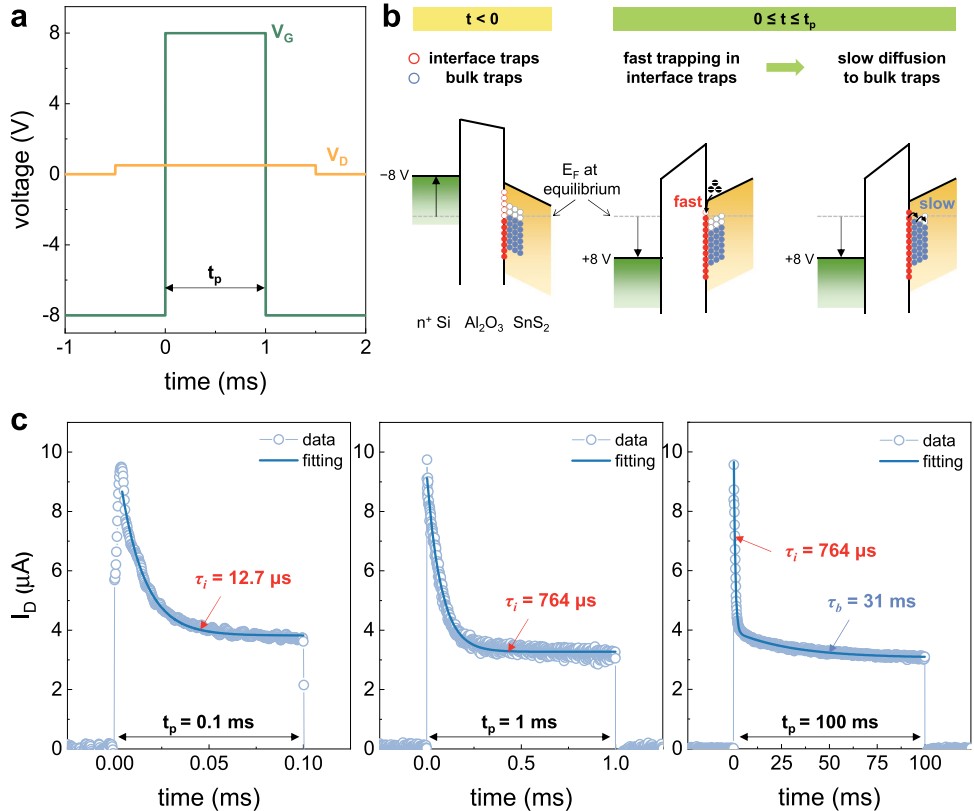

**Fig. 3 Analysis of different trap time constants. a** Schematic diagram of gate and drain pulses applied to the SnS$_2$ memtransistor for measuring $I_D$ transient. **b** Energy band diagram schematics showing the key mechanism responsible for the resistive switching. **c** $I_D$ transient results according to differing gate pulse durations ($t_p$). Obtained short trap time constant ($\tau_i$) below 764 μs is responsible for the interface trap. Meanwhile, a trap time constant ($\tau_b$) longer than tens of milliseconds is responsible for the bulk trap.

where $I_0$ is the steady-state drain current, $A$ is a fitting parameter, and $\tau_i$ is the trap time constant for the interface trap. As $t_p$ increases from 0.1 to 1 ms, $\tau_i$ increases to 764 μs because additional interface traps contribute to the $I_D$ transient. However, when $t_p$ is >10 ms, the $I_D$ curve can only be fitted using two different trap time constants, thus different types of traps additionally contribute to the $I_D$ transient. $I_D$ is expressed as $I_D = I_0 + A \cdot \exp(-t/\tau_i) + B \cdot \exp(-t/\tau_b)$, where $B$ is another fitting parameter and $\tau_b$ is the trap time constant for the bulk trap. Note that the extracted $\tau_i$ was fixed at 764 μs, but the extracted $\tau_b$ was significantly longer than $\tau_i$ (several tens of milliseconds). This longer $\tau_b$ was owing to the slow electron emission process from the bulk trap. As aforementioned, the resistive switching behavior of our SnS$_2$ memtransistor is only observed when the duration of the gate pulse is longer than 5 ms. Therefore, it can be concluded that the trapped charges at the bulk traps inside the SnS$_2$ layer result in reliable nonvolatile resistive switching of the SnS$_2$ memtransistor, which can be exploited for the analog–digital hybrid computing, as discussed below.

**Continuous-time Kalman filter algorithm and its analog Kalman filter circuit.** As mentioned in above, the traditional Kalman filter algorithm is expressed in a discrete-time form such that it can be executed in a conventional digital processor, and it should then be converted into a continuous-time form to be executed in the proposed memtransistor-based analog circuit. From the continuous-time Kalman filter theory[24] (see the details in Supplementary Note 5–1), the update equation for Euler angles ($E(t)$) can be written as

$$\dot{E}_{est}(t) = \varpi_{mea}(t) + K[E_{mea}(t) - E_{est}(t)]. \quad (2)$$

In detail, in our experiment, the roll ($\phi$) and pitch ($\theta$) angles oscillated in the range of ±30°; therefore, $E(t)$ represents $\phi(t)$ or $\theta(t)$. As shown in Fig. 1c, the measured raw data from the sensor module ($p$, $q$, $r$, $A_x$, $A_y$, and $A_z$) were delivered to the microcontroller. The microcontroller then calculated Eq. (S1), and subsequently generated an $\varpi_{mea}(t)$ signal using a digital-to-analog converter (DAC), thus $\varpi_{mea}(t)$ implies the angular velocity obtained by the gyroscope ($\varpi_{mea}(t) = \dot{\phi}_{mea}(t)$ or $\dot{\theta}_{mea}(t)$). Similarly, the microcontroller calculated using Eq. (S5) generated the $E_{mea}(t)$ signal. $E_{mea}(t)$ implies the Euler angles obtained by the accelerometer ($E_{mea}(t) = \phi_{mea}(t)$ or $\theta_{mea}(t)$). In addition, $E_{est}(t)$ is the output of the Kalman filter ($\phi_{est}(t)$ or $\theta_{est}(t)$) and $K$ is the Kalman gain. Figure 4a shows the transfer function block diagram representing Eq. (2) in the frequency domain. Figure 4b shows the circuit diagram of the analog Kalman filter for implementing Eq. (2). Note that the performance of our analog Kalman filter depends on the $K$ value, which is determined by the conductance of the memtransistor ($G_K$), where $K$ is determined by $K = G_K/C_f$ (see details in Supplementary Note 5–2).

Figure 4c shows the input signals of the analog Kalman filter circuit for the roll angle, that is, $\phi_{mea}(t)$ (green curve) and $\dot{\phi}_{mea}(t)$ (orange curve). Owing to the limited operating voltage range of the operational amplifier, the microcontroller generated input signals such that 0.1 V of a signal level represents an angle of 1 degree. Consequently, a rotation of ±30° in the roll angle is represented as a signal in the range of ±3 V. Note that the $\phi_{mea}(t)$ obtained by the accelerometer has high-frequency noise, and $\dot{\phi}_{mea}(t)$ obtained by the gyroscope overestimates the actual rotation angle. As previously explained, both the gyroscope and

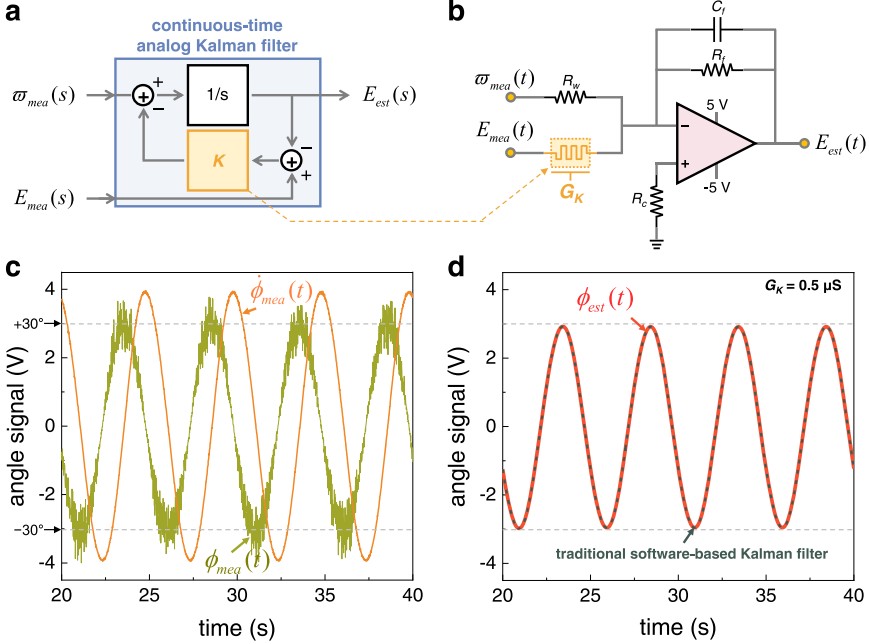

**Fig. 4 Continuous-time analog Kalman filter circuit. a** Block diagram of the signal filtering (Euler angle estimation) with continuous-time analog Kalman filter. The Laplace transform of the complex frequency is denoted by $S$. The Euler angle ($E_{est}(s)$) can be estimated through the continuous-time analog Kalman filter algorithm, where $\varpi_{mea}(s)$ signal obtained by the gyroscope and $E_{mea}(s)$ signal obtained by the accelerometer are generated from the microcontroller. **b** Circuit schematic diagram of the memtransistor-based analog Kalman filter, where the Kalman gain is reconfigurable by adjusting the conductance of the memtransistor (i.e., $K = G_K/C_f$). In our experiment, $G_K = 0.5\ \mu S$, $R_w = 100\ k\Omega$, $R_f = 700\ k\Omega$, $R_c = 100\ k\Omega$, and $C_f = 10\ \mu F$ were used. **c** Two input signals ($\phi_{mea}(t)$ and $\dot{\phi}_{mea}(t)$) for the roll angle. **d** Filtered output (estimated roll angle, $\phi_{est}(t)$) by the continuous-time analog Kalman filter circuit. The output accurately estimated the roll angle oscillation (i.e., ±30°), and it shows a good agreement with the result obtained by the traditional software-based Kalman filter.

accelerometer cannot be used alone to accurately estimate Euler angles. However, the signal filtered by our analog Kalman filter (red curve in Fig. 4d) can effectively eliminate the noise and estimate the roll angle accurately, which demonstrates the feasibility of sensor fusion through our analog Kalman filter. The same result was obtained for pitch angle, as shown in Fig. S12. Moreover, the output of our memtransistor-based analog Kalman filter shows good agreement with the result of the traditional software-based Kalman filter (black dotted curve in Fig. 4d), which was calculated entirely on the microcontroller without using the memtransistor. The consistency of these results with/without the memtransistor shown in Fig. 4d guarantees the feasibility of our analog Kalman filter circuit. In addition, according to Eq. (2), the performance of the analog Kalman filter depends on $G_K$ (Fig. S13); therefore, the $G_K$ should be optimized to achieve the best Kalman filtering performance. The precise tunability of the $G_K$ in our SnS₂ memtransistor can provide reliable reconfigurability for optimizing the performance of our analog Kalman filter circuit.

**Comparative analysis of the power consumption**. We must ascertain whether our analog−digital hybrid computing is more energy efficient than traditional digital computing for sensor fusion applications. A current waveform analyzer was used for quantitative power consumption analysis in our study (Fig. S14a and see Supplementary Note 6), which enables real-time monitoring of the current flow and subsequent calculation of the power consumption. As explained in earlier, our analog−digital hybrid computing platform is composed of a digital component (the microcontroller) and an analog component (the memtransistor-based analog Kalman filter circuit). The power consumption of the digital component ($P_d$) can be obtained by

measuring the voltage ($V^+$) and current ($I_d$) supplied to the microcontroller (Fig. S14b). Similarly, the power consumption of the analog component ($P_a$) can be obtained by measuring the supply voltages ($V^+$ and $V^−$), supply current ($I_a$), and output voltage and current ($V_o$ and $I_o$). In the case of the traditional software-based Kalman filter, the entire algorithm is executed only on the microcontroller; therefore, the power consumption required for the software-based Kalman filter ($P_{soft-K}$) can be evaluated using $P_d$, that is, $P_{soft-K} = P_d$. Meanwhile, in the case of our hybrid computing-based Kalman filter, the algorithm is calculated through an analog circuit, but the conversion of the sensing signal is responsible for the digital component using the DAC. Therefore, the total power consumption required for our hybrid computing-based Kalman filter ($P_{hybrid-K}$) is the summation of $P_d$ and $P_a$, that is, $P_{hybrid-K} = P_d + P_a$.

It is necessary to establish a power-consumption criterion for comparison. In our experiment, because the roll and pitch angles oscillated repeatedly, we assigned the total amount of power consumption during one period of angle oscillation (±30° change in $\phi$ and $\theta$ for 5 s, as shown in Fig. S14c) as comparison criteria. Fig. S15 shows the summarized results of evaluating the power consumption. In the case of the traditional software-based Kalman filter (Fig. S15a), the total power consumption during one period of angle oscillation was 197 mJ. Meanwhile, in the case of our hybrid computing-based Kalman filter (Fig. S15b), the analog component consumed only 0.79 mJ. The power consumption of the digital component was also reduced to 53.7 mJ owing to the analog circuit mitigating the computational burden of the algorithm calculation. Consequently, our analog-digital hybrid computing platform can implement reliable sensor fusion with only 1/4th of the power consumption compared to the traditional software-based method.

## Discussion

In this study, an analog–digital hybrid computing platform was demonstrated for low-power and high-accuracy sensor fusion in drones. The hybrid computing platform was built by the co-integration of the conventional digital processor (digital component) and $SnS_2$ memtransistor-based analog circuit (analog component). The developed continuous-time Kalman filter algorithm was implemented using an analog Kalman filter circuit, where the precise tunability and high reliability of the $SnS_2$ memtransistor enabled stable algorithm execution. The output of the analog Kalman filter circuit shows good agreement with the result of a traditional software-based Kalman filter, which provides a practical accuracy of analog-digital hybrid computing. In addition, our hybrid computing platform achieves higher energy efficiency while mitigating the burden on the battery capacity and computing power of the digital component. Notably, because the memtransistor does not require to be reprogrammed frequently and is non-volatile, the power consumption required to optimize the performance of the computing platform can be minimized. Consequently, the memtransistor-based analog–digital hybrid computing platform can be applied to a broad spectrum of applications that require processing sensing data with limited energy, such as the Internet of Things and edge computing applications.

We believe that the power consumption of our hybrid computing platform can be further reduced when the sensor module is integrated directly into the analog Kalman filter circuit. Because the measured sensing signal is analog, the analog Kalman filter circuit can process the data directly without using any analog-to-digital conversion, thereby minimizing both the latency and quantization error. In addition, because the memtransistor has one more electrode (gate electrode) than a conventional two-terminal memristor, the device conductance can be adjusted through the gate electrode, which enables simpler circuit configuration (see Supplementary Note 7 and Fig. S16). A simple circuit configuration is also expected to contribute in reducing the overall power consumption.

With regards to similar previous report[21], our analog component can provide higher performance and reliability; our $SnS_2$ memtransistor showed higher endurance (above $10^5$) with lower operation current level (about 1 μA). In addition, the output of our memtransistor-based analog Kalman filter was comparatively analyzed with the result from a traditional software-calculated Kalman filter. This comparative study clearly guarantees the feasibility of our analog-digital hybrid computing. Moreover, our study mainly focused on improving the accuracy and energy efficiency in sensor fusion through analog-digital hybrid computing, which is completely different from the previous report[21] that mainly focused on the improvement of speed (response time) in robot control.

## Methods

**Electrical measurement**. The electrical pulses ($V_G$ and $V_D$) were generated by a function generator (Keysight 33622a) and drain current ($I_D$) was measured by a source-measurement unit (SMU, Keysight B2902a). The pulse was applied to the gate and drain electrodes, and the drain current was measured through the source electrode. Additionally, to evaluate the power consumption in real-time, a current waveform analyzer (Keysight, CX3300) was used to monitor the amount of current flow to both the digital and analog components.

**Fabrication of the $SnS_2$ memtransistor**. The $SnS_2$ memtransistor, which served as the gate electrode, was fabricated on a heavily n-doped Si wafer with a resistivity of <0.005 $\Omega \cdot$ cm (QL Electronics Co.). A 20-nm-thick aluminum oxide ($Al_2O_3$) film, which is a gate dielectric layer, was grown via atomic layer deposition (Nano-ALD2000, IPS) at 350 °C. In the next step, we obtained a thin flake of $SnS_2$ from bulk $SnS_2$ (HQ Graphene) by mechanical exfoliation using the scotch tape method and transferred it gently on top of a polydimethylsiloxane (PDMS) stamp. We selected the desired flake on PDMS through an optical microscope and transferred

it onto a Si/$Al_2O_3$ substrate (dry transfer method) through a micromanipulator. Finally, the source and drain electrodes were patterned using electron beam lithography. Ti/Au (10 nm/50 nm) metals were deposited using a thermal evaporator at high vacuum pressure (~$10^6$ Torr) and patterned using a conventional lift-off process.

## Data availability

The authors declare that the main data supporting the findings of this study are available within the paper and its Supplementary Information files. Source data are provided with this paper.

## Code availability

The code presented and used in this publication is available from the corresponding authors on reasonable request.

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

## Acknowledgements

This research was supported by the National Research Foundation of Korea (NRF) grants funded by the Ministry of Science and ICT (2019R1A2C1002491 and 2020R1A6A1A03038540).

## Author contributions

S.K. supervised the project overall. S.R. and M.F.K. performed the device fabrication and the basic electrical measurement. H.D.K. supports TEM and EDX analysis. S.K. performed the circuit integration, the code simulation, and the electrical analysis. S.K. involved in the writing, review, and editing of the manuscript.

## Competing interests

The authors declare no competing interests.
