## [Peer Review File · Nature Communications]

REVIEWER COMMENTS

Reviewer #1 (Remarks to the Author):

1. It is not clear what is the main goal in this application of using an analog kalman filter. Is stability an issue and you require computational speed? Is it just the power? Something else? Please highlight and clarify better.
2. What is the advantage of using a memtransistor over a memristor? What is exactly the advantage in this application and in real-time robot control?
3. It is not clear how the value of the device conductance (the Kalman gain) was set. What defined it and allowed you to keep it at a constant value. Please clarify further this process.
4. Provide please more details about the actual effect of the noise in the control of the robot. Provide quantitative Information. Please also quantify the levels of the noise for the different sensor signals, the SNR and the effect on the robot operation.
5. Why not all the computation or more of the computation being performed in the analog domain? Please comment and provide a vision for the next steps. Page 4 of the paper line 11, you introduce too suddenly the concept of the hybrid computing approach.
6. Please provide more information on how the conductance is being measured, what signals, values, apparatus, setup, etc.
7. What chip did you use for the amplifier and how did you select the values for the other components?
8. Please explain why a memtransistor was not use for the second signal.
9. How is the transfer of the SnS2 layer is performed, what is the throughput and repeatability of the process? How well defined are the device parameters using this process?
10. Why haven't you made a proper PCB and a breadboard is being used?
11. In my opinion the main paper should show some results from the operation and control of the robot with and without the memtransistors and show some results demonstrating the usefulness of the proposed approach, rather than having all results in the SI.
12. With regards to ref 21, the main differences are that yo used a memtransistor as opposed to a memristor and you only considered the power consumption, when these other authors also considered the computational speed and its effect on robot control. Both used Kalman filtering and fusion. Consequently for at this point the innovation does not appear to be significant. Please highlight further what your innovation is, especially when compared to that other paper.

Reviewer #2 (Remarks to the Author):

The authors reported a hybrid circuit of sensor fusion for the rotation motion of a drone. Most importantly, they demonstrated the Kalman filter using SnS2 "memristor" based analogue circuits. They also showed this approach has lower latency and is more energy efficient. This is a very good work, however, there are still a few issues that need to be addressed.

- 1) Memristor is a 2-terminal device. The SnS2 device is actually a 3-terminal device, not really a memristor. Of course, they used the device like a 2-terminal memristor with a fixed gate voltage. To prevent confusion, it would be nice to clarify this.
- 2) To implement the Kalman filter properly, the resistance of each "memristor" needs to be

independent of the voltage on the device. Therefore, a linear I-V curve is preferred. Yes, they included the IV curves at fix gate voltages. But the Kalman filter is worked with $V_g=0$, and the $V_g=0$ curve is too hard to see. A zoom-in view of $V_g=0$ curve is needed.

3) If the authors could compare this SnS2 device with other types of memristors and explain why this device is used, it would be useful.

4) In the paper, the system is built on bread boards. It is a good concept demonstration; however, it is too big/heavy for drone. If they can put the system on a PCB, it would be nice.

Responses to the editor and the reviewers

We are very grateful to the editor and reviewers for the positive, constructive, and detailed comments on our previous manuscript (NCOMMS-22-07830). We have made relevant revisions in response to these valuable comments. The list of changes in the revised manuscript and the point-by-point responses are detailed below. Our responses are indicated in blue colored text, and the additions (or revisions) to the manuscript are marked in red colored text. We believe that the revised version is now suitable for publication in *Nature Communications*.

Reviewer #1:

Response: We appreciate the careful and constructive comments received from the reviewer. We have provided point-by-point responses for the reviewer's comments below. Our responses are indicated in blue colored text, and the additions (or revisions) to the manuscript are marked in red colored text.

1) It is not clear what is the main goal in this application of using an analog Kalman filter. Is stability an issue and you require computational speed? Is it just the power? Something else? Please highlight and clarify better.

Response: Thank you for your queries. The primary objective of our study is to implement a "low-power computing" using an analog-digital hybrid circuit system.

In the case of mobile robots, drones, or IoT devices, power consumption is a critical issue because their battery capacity is limited. Particularly in the case of drones, more sensors and advanced machine learning algorithms are being integrated to perform more complex missions. However, a high-performance processor that can simultaneously process multiple sensing data inevitably consumes a lot of energy, which shortens the flight time.

At the beginning of our study, we assumed that if computational burden of the digital processor can be partially distributed to the analog circuit, the overall power consumption could be reduced. To validate this, a drone was selected as an application target in this study. From the various functions performed by the drone's digital processor, sensor fusion was implemented through our memtransistor-based analog circuit. Because the Kalman filter is the most widely used algorithm in drones for sensor fusion, we implemented an analog Kalman filter circuit in this study.

To clarify the main goal of our study, we revised the INTRODUCTION section as below.

Revised INTRODUCTION (last paragraph): In this study, we demonstrate a memtransistor (memristor with transistor structure²²)-based analog-digital hybrid computing platform for sensor fusion with higher energy efficiency. The measured data from both the gyroscope and accelerometer were combined to accurately determine the Euler angles of drones, wherein the Kalman filter algorithm was implemented using a customized analog circuit with the memtransistor. Because this analog Kalman filter circuit can operate independently without using the computing resources of the microcontroller, the computational burden on the microcontroller is reduced, and subsequently a reduction in overall power consumption can be expected. Here, we exploited transition-metal dichalcogenide (TMD) materials to implement the memtransistor, where the bulk traps located at the tin disulfide (SnS₂)-aluminum oxide (Al₂O₃) stack provide a highly reliable nonvolatile resistive switching

behavior. The precise tunability of the SnS₂ memtransistor allows for the reconfigurability of our analog-digital hybrid computing platform. Finally, we experimentally demonstrated that a drone using our hybrid computing performs sensor fusion with higher energy efficiency than a drone with only a conventional digital processor.

2) What is the advantage of using a memtransistor over a memristor? What is exactly the advantage in this application and in real-time robot control?

Response: Thank you for your queries. Because the memtransistor can control the device conductance by using the gate electrode, the peripheral circuit configuration for the memtransistor can be simplified more compared to that for the memristor.

Figure R1. (a) Memristor with two electrodes. (b) Example analog circuit with a memristor. (c) Memtransistor with three electrodes. (d) Example analog circuit with a memtransistor.

A memristor has only two electrodes, thereby V_{write} (i.e., voltages for adjusting the conductance) must be

applied to both ends of the two electrodes (Fig. R1a). However, these electrodes are also used for signal input/output. Consequently, additional peripheral circuits are required to distribute the input signal ($E_{mea}(t)$) and V_{write} properly (Fig. R1b).

In contrast, a memtransistor has three electrodes, and the conductance can be adjusted by using the gate electrode independently (Fig. R1c). Therefore, the conductance can be adjusted without additional peripheral circuits (Fig. R1d). In summary, there is no difference between the memristor and memtransistor in the conductance change characteristic itself, but the memtransistor enables simpler circuit configuration.

We have revised CONCLUSION section to clarify the difference between a memristor and memtransistor.

Revised CONCLUSION (last paragraph): We believe that the power consumption of our hybrid computing platform can be further reduced when the sensor module is integrated directly into the analog Kalman filter circuit. Because the measured sensing signal is analog, the analog Kalman filter circuit can process the data directly without using any analog-to-digital conversion, thereby minimizing both the latency and quantization error. In addition, because the memtransistor has one more electrode (gate electrode) than a conventional two-terminal memristor, the device conductance can be adjusted through the gate electrode, which enables simpler circuit configuration. A simple circuit configuration is also expected to contribute in reducing the overall power consumption.

3) It is not clear how the value of the device conductance (the Kalman gain) was set. What defined it and allowed you to keep it at a constant value. Please clarify further this process.

Response: Thank you for your comment. As shown in Fig. S12, the performance of the analog Kalman filter depends on the Kalman gain value. The Kalman gain (K), where the output of analog Kalman filter (i.e., angle signal) could be ± 3 V, was determined using a binary search method (trial and error). When K is 0.05 (Fig. S12), the angle signal has an amplitude of exactly ± 3 V, which means that the analog Kalman filter accurately estimates the actual range of Euler angle oscillation ($\pm 30^\circ$). Thereafter, as discussed in Supporting Note 5-2, K is determined by $K = G_K/C_f$, where C_f was 10 μF in our study. Consequently, the optimized device conductance (G) was 0.5 μS .

Next, as discussed in Supporting Note 4-3, the device conductance was adjusted by using the update-verify feedback method. This method is a well-known technique for accurately adjusting the conductance of a memristor [F. Alibart et al., *Nanotechnology*, 23, 075201, 2012]. We also demonstrated the system for this method in our previous work [S. Kim et al., *Nanoscale*, 11, 21449-21457, 2019]. Fig. R2 shows the flowchart

for the update-verify feedback method. A detailed description of the feedback process had been added in the Supporting Note 4-3 as follows:

Revised Supporting Note 4-3: Fig. S7 shows a flow chart for the update-verify process for the precise control of device conductance (G). The following process is repeated until the desired G is obtained.

- (1) At the outset, G_{target} and a threshold value must be defined. G_{target} is the desired G value. The threshold value is the acceptable limit of the relative error between G and G_{target} . In our experiment, we set the threshold value as $\pm 5\%$.
- (2) G is measured using a read pulse ($V_G = 0$ V, $V_D = 0.5$ V, 50 ms) and the relative error (G_{var}) is calculated.
- (3) If G_{var} is within the predefined threshold (i.e., $\pm 5\%$), the feedback process is considered complete and the process is stopped.
- (4) Else, action is taken based on the sign of G_{var} . For negative G_{var} , a potentiation pulse ($V_G = -10$ V, $V_S = V_D = 0$ V, 50 ms) is applied to increase G . Meanwhile, for positive G_{var} , a depression pulse ($V_G = +8$ V, $V_S = V_D = 0$ V, 50 ms) is applied to decrease G .

In the experimental demonstration (Fig. 2e), approximately 10 repeated feedback processes are required to achieve the desired G value.

4) Provide please more details about the actual effect of the noise in the control of the robot. Provide quantitative Information. Please also quantify the levels of the noise for the different sensor signals, the SNR and the effect on the robot operation.

Response: Thank you for your comment. You have requested additional explanation about the effect of noise on "the control of the robot," but our study is focused only on "the sensor fusion in drones." Therefore, we would like to focus our answer only on the effect of the sensor's noise in the operation of drones.

First, the quantitative noise performance of the sensors used in our study is as follows. In our experiment, a commercial IMU sensor (MPU6050, Supporting Note 2-1) was used because it is commonly employed in small drones. From the datasheet of MPU6050, the power spectral density of the accelerometer noise is $400 \mu\text{g}/\sqrt{\text{Hz}}$ (at 10 Hz). Additionally, the power spectral density of the gyroscope noise is $0.005 \text{ }^\circ/\text{s}\cdot\sqrt{\text{Hz}}$ (at 10 Hz). We have added this information to Supporting Note 2-1.

Furthermore, we have already discussed the effect of sensor noise in Supporting Note 2-2.

Figure R2. Data to show the noise effect on the Euler angle estimation (a copy of Fig. S2).

Fig. R2a and Fig. R2b show the measured raw data of angular velocities (p , q , and r) and accelerations (A_x , A_y , and A_z), respectively. Note that the measured angular velocities (Fig. R2a) do not contain any noticeable noise but the measured accelerations (Fig. R2b) contain a high frequency noise. The effect of a sensor's noise becomes more obvious after estimating the Euler angle. Fig. R2c shows the Euler angles calculated using both Eq. (S1) and the measured angular velocities from the gyroscope. Although the estimated Euler angles do not contain any noticeable high-frequency noise, a drift (i.e., a gradual change with the time) is clearly observed. In contrast, Fig. R2d shows the Euler angles calculated using both Eq. (S5) and the accelerations measured through the accelerometer. The estimated Euler angles shown in Fig. R2d evidently have a high frequency noise and they overestimate the drone's actual oscillation range ($\pm 30^\circ$). Therefore, because of the intrinsic noise from the gyroscope and accelerometer, the Euler angles cannot be estimated accurately without the sensor fusion.

We have added these explanations to Supporting Note 2-2 as follows:

Revised Supporting Note 2-2: Fig. S2a and Fig. S2b show the measured raw data of angular velocities (p , q , and r) and accelerations (A_x , A_y , and A_z), respectively. **Note that the measured angular velocities do not contain any noticeable noise but the measured accelerations contain a high frequency noise.** The effect of a sensor's

noise becomes more obvious after estimating the Euler angles. Using Eq. (S1) and Eq. (S5), the roll and pitch angles can be calculated (Fig. S2c and Fig. S2d). When the data of only angular velocities are used (Fig. S2c), the estimated roll and pitch angles are sufficiently consistent with the drone's actual oscillation (the amplitude of $\pm 30^\circ$). However, a drift in which the error gradually accumulates over time is observed. Conversely, when the data of accelerations are only used (Fig. S2d), there is no drift, but the estimated roll and pitch angles include a high frequency noise. In addition, the oscillation range is also overestimated. Therefore, due to an intrinsic bias instability from the gyroscope and a high frequency noise from the accelerometer, Euler angles cannot be estimated accurately without exploiting the sensor fusion.

5) Why not all the computation or more of the computation being performed in the analog domain? Please comment and provide a vision for the next steps. Page 4 of the paper line 11, you introduce too suddenly the concept of the hybrid computing approach.

Response: Thank you for your comment. As already mentioned in the INTRODUCTION section, an analog-digital hybrid computing platform is typically referred to as neuromorphic system. The basic concept of the neuromorphic system was proposed in the 1990s [Ref. 8]. After the first development of an analog resistive switch (i.e., memristor) in 2008 [D. B. Strukov et al., Nature, 453, 80-83, 2008], the research of neuromorphic systems began to attract a great attention.

The ultimate goal of the neuromorphic system research is to implement an energy-efficient computing system that can overcome the limitations of the conventional digital processor by mimicking the advantages/features of biological neural networks (i.e., analog and massive parallelism). However, because the operating principles of neural networks (mechanisms of learning and cognitive processes) have not yet been clearly understood, it is currently impossible to demonstrate a neuromorphic system based on an artificial neural network that can replace all functions of conventional digital processors. Therefore, the research on neuromorphic systems currently in progress has been mainly focused on alleviating the burden of complex calculations (especially in machine learning) by using crossbar arrays of memristors [Refs. 9–16]. A key machine learning algorithm heavily relies on matrix vector multiplications (MVM), wherein high-density crossbar array of memristors is well-suited to accelerate such algebraic operations.

The analog-digital hybrid circuit studied in our work is also a branch of the neuromorphic system research. The analog switching behavior of the memristor is useful for energy-efficient analog computing and enables functional reconfigurability of the circuit. There are several proposals and experimental demonstrations of simple circuits exploiting the analog properties of memristors, e.g., tunable gain in operating amplifiers [R. Berdan et al., Appl. Phys. Lett., 101, 243502, 2012], analog-to-digital (ADC) and digital-to-analog (DAC)

circuits [L. Gao et al., 2013 NANOARCH], finite impulse response (FIR) filter [Y. Hong et al., IEEE Trans. Circuits and Systems, 62, 1392-1401, 2015], etc.

Our research does not aim to implement a computing system that can process all data analogously like the human brain. On the contrary, our goal is to realize an analog-digital hybrid computing system in which analog circuits can reduce the overall power consumption while maintaining the performance of the digital processor (i.e., high arithmetic accuracy and speed). In our next study, we will try to experimentally demonstrate more complex computations than sensor fusion by implementing an advanced analog-digital hybrid circuit.

We revised INTRODUCTION to clarify the purpose of the analog-digital hybrid computing platform as follows:

Revised INTRODUCTION (4th paragraph): Notably, an analog-digital hybrid computing platform, which is inspired by biological neural networks (typically referred to as neuromorphic systems⁸), has been considered as a promising candidate for realizing energy-efficient computing.⁹⁻¹³ The precisely tunable analog resistive switch (i.e., memristor) enables **energy-efficient analog computation with a process-in-memory architecture and also allows for functional reconfigurability. The feasibility of the analog-digital hybrid computing platform** has been successfully demonstrated to mitigate the computational burden of vector-matrix multiplication in the calculation of various machine-learning algorithms.¹⁴⁻¹⁶ **The research to reduce overall energy consumption by replacing a part of digital calculation with analog circuits is being conducted in various application fields.** Furthermore, recent advancements in memristors based on two-dimensional materials offer the possibility of designing new materials with atomic-level precision, resulting in excellent resistive switching performance with only a small amount of energy consumption.¹⁷⁻²⁰ A drone is a complex real-time sensing system that can benefit substantially from this memristor-based analog-digital hybrid computing platform. Because the Kalman filter algorithm can be expressed by linear equations, it can be implemented using memristor-based analog circuits. Moreover, this memristor-based analog component can operate independently without using computing resources from the digital processor, thereby reducing the computational load of the digital component. Nevertheless, there exist some demonstrations of memristor-based hybrid computing for drones but only one recent study has applied this to control an inverted pendulum for a mobile robot.²¹

6) Please provide more information on how the conductance is being measured, what signals, values, apparatus, setup, etc.

Response: Thank you for your comment. As shown in Fig. R3, the electrical pulses (V_G and V_D) were generated

by a function generator (Keysight 33622a) and drain current (I_D) was measured by a source-measurement unit (SMU, Keysight B2902a). The pulse was applied to the gate and drain electrodes, and the drain current was measured through the source electrode.

Figure R3. Setup for the characterization of our memtransistor's electrical performance.

The device conductance was measured through a read pulse ($V_G = 0$ V, $V_D = 0.5$ V, 50 ms), as shown in Fig. 2d. The device conductance (G) was then calculated by $G = I_D/V_D$. We confirmed that the read pulse is nondestructive; hence, the device conductance cannot be changed by the read pulse.

We add this information to the METHODS section.

Added to METHODS: The electrical pulses (V_G and V_D) were generated by a function generator (Keysight 33622a) and drain current (I_D) was measured by a source-measurement unit (SMU, Keysight B2902a). The pulse was applied to the gate and drain electrodes, and the drain current was measured through the source electrode. Additionally, to evaluate the power consumption in real-time, a current waveform analyzer (Keysight, CX3300) was used to monitor the amount of current flow to both the digital and analog components.

7) What chip did you use for the amplifier and how did you select the values for the other components?

Response:

Figure R4. Analog Kalman filter circuit, which is a revised version of Fig. S10. In our experiment, $R_w = 100$ k Ω , $R_f = 700$ k Ω , $R_c = 100$ k Ω , and $C_f = 10$ μ F.

Thank you for your question. We used an operational amplifier (op-amp, Linear Technology, LT6005) for our analog Kalman filter because it is specially designed for low-power applications (Fig. R4). This op-amp can operate on power supplies as low as 1.6 V while only drawing a maximum of 1 μ A quiescent current.

As discussed in Supporting Note 5-2, when the Kalman gain value (K) is initially set, R_w and C_f values are automatically determined by Eq. (S25). In addition, because R_c does not affect the output of the circuit according to Eq. (S23), R_c was set arbitrarily as 100 k Ω . Finally, R_f was optimized through circuit simulation.

We revised the caption of Fig. S10 to indicate the model of the op-amp and explain how to determine the value of components.

Revised caption of Fig. S10: The analog Kalman filter circuit (a copy of Fig. 4b). We used an operational amplifier (Linear Technology, LT6005) for the circuit configuration. When the Kalman gain (K) is initially set, R_w and C_f are automatically determined by Eq. (S25). Because R_c does not affect the output of the circuit according to Eq. (S23), R_c was set arbitrarily as 100 k Ω . R_f was optimized through circuit simulation. Consequently, $R_w = 100$ k Ω , $R_f = 700$ k Ω , $R_c = 100$ k Ω , and $C_f = 10$ μ F.

8) Please explain why a memtransistor was not use for the second signal.

Response: Thank you for your comment. As pointed out by the reviewer, two memtransistors are actually required for each input signal ($E_{mea}(t)$ and $\varpi_{mea}(t)$) to fuse the signals according to the Kalman gain.

However, in our experiment, one memtransistor was intentionally utilized for an easier understanding of the sensor fusion algorithm (i.e., Kalman filtering used in this study).

In the Kalman filtering process, there is only one variable that needs to be continuously updated, i.e., the Kalman gain. Therefore, we corresponded one Kalman gain value to one memtransistor conductance. Although G_K , R_w , and C_f contribute to determining the Kalman gain value in the analog Kalman filter circuit, we used a single memtransistor to effectively convey our idea that the Kalman gain can be stored in a non-volatile memtransistor.

9) How is the transfer of the SnS₂ layer is performed, what is the throughput and repeatability of the process? How well defined are the device parameters using this process?

Response: Thank you for your queries. First, we added the following explanations about the transfer method of the SnS₂ layer to the METHODS section:

Revised METHODS section: The SnS₂ memtransistor, which served as the gate electrode, was fabricated on a heavily n-doped Si wafer with a resistivity of less than 0.005 Ω·cm (QL Electronics Co.). A 20-nm-thick aluminum oxide (Al₂O₃) film, which is a gate dielectric layer, was grown via atomic layer deposition (NanoALD2000, IPS) at 350 °C. **In the next step, we obtained a thin flake of SnS₂ from bulk SnS₂ (HQ Graphene) by mechanical exfoliation using the scotch tape method and transferred it gently on top of a polydimethylsiloxane (PDMS) stamp. We selected the desired flake on PDMS through an optical microscope and transferred it onto a Si/Al₂O₃ substrate (dry transfer method) through a micromanipulator.** Finally, the source and drain electrodes were patterned using electron beam lithography. Ti/Au (10 nm/50 nm) metals were deposited using a thermal evaporator at high vacuum pressure (~10⁶ Torr) and patterned using a conventional lift-off process.

Next, Fig. R5 shows the measured I_D-V_G curves from 6 devices fabricated using the same process. There is a slight device-to-device variation. Because the transfer method of the SnS₂ layer is a manual process, we believe that that the variation is owing to the thickness variation of the SnS₂ layer for each device. Nevertheless, this variation is not an issue in the operation of our analog-digital hybrid computing platform. Regardless of the device variation, the desired channel conductance can be precisely adjusted through the update-verify feedback method discussed in Supporting Note 4-3.

Figure R5. Measured I_D - V_G curves from 6 different devices.

10) Why haven't you made a proper PCB and a breadboard is being used?

Response: Thank you for your question. In this work, for the proof-of-the concept of our analog-digital computing platform, we implemented the analog circuit on a breadboard.

In fact, to integrate our analog Kalman filter circuit on a PCB and mount it on the drone, 1) a circuit for generating a voltage pulse, 2) a circuit for performing the update-verify feedback process, and 3) a circuit for power distribution are additionally required. Therefore, further research on the design of these peripheral circuits is necessary. We hope that the system-level analog-digital hybrid computing platform will be demonstrated in a near-future study.

11) In my opinion the main paper should show some results from the operation and control of the robot with and without the memtransistors and show some results demonstrating the usefulness of the proposed approach, rather than having all results in the SI.

Response: Thank you for your comment. We have shown the comparative results with and without a memtransistor in Fig. 4d. Fig. 4d shows the output of our memtransistor-based analog Kalman filter (red curve). Note that the black dotted curve denotes the output of the traditional Kalman filter, which is a purely software-calculated result obtained by the microcontroller without using any memtransistor. It is obvious that both results

show a good consistency, which guarantees the feasibility of our analog Kalman filter circuit.

We have added further explanations to section 3.1 clarify the comparative results with/without memtransistor as follows:

Revised Section 3.2 (pages 11–12): Moreover, the output of our memtransistor-based analog Kalman filter shows good agreement with the result of the traditional software-based Kalman filter (black dotted curve in Fig. 4d), which was calculated entirely on the microcontroller without using the memtransistor. The consistency of these results with/without the memtransistor shown in Fig. 4d guarantees the feasibility of our analog Kalman filter circuit.

12) With regards to ref 21, the main differences are that you used a memtransistor as opposed to a memristor and you only considered the power consumption, when these other authors also considered the computational speed and its effect on robot control. Both used Kalman filtering and fusion. Consequently, for at this point the innovation does not appear to be significant. Please highlight further what your innovation is, especially when compared to that other paper.

Response: Thank you for your comment. The differences between our study and Ref. [21] are as follows.

< Comparison of the device >

- 1) Ref. [21] utilized a two-terminal memristor, but our work utilized a three-terminal memtransistor. As already answered in Question#2, the memtransistor enables a simpler circuit configuration.
- 2) Ref. [21] used a Pt/Al₂O₃/Ta/Pt memristor device and this metal-oxide based memristor has already been studied in many previous works. In contrast, we demonstrated a reliable resistive switching characteristic using a new material (SnS₂ nanosheet) and analyzed its physical mechanism.
- 3) The performance of resistive switching obtained from our SnS₂ memtransistor is better than that of the memristor in Ref. [21]. The endurance of our SnS₂ memtransistor is above 10⁵ (Fig. 2d), but that of the memristor in Ref. [21] is approximately 5,000. Additionally, the operation current level of our SnS₂ memtransistor during the resistive switching is under 1 μA (Fig. 2d), but that of the memristor in Ref. [21] is over 100 μA. Therefore, our SnS₂ memtransistor is better suited for low-power operation.

< Comparison of the sensor fusion >

- 1) As answered in Question#11, in our study, the output of our memtransistor-based analog Kalman filter

is comparatively analyzed with the result from a traditional software-calculated Kalman filter. This comparative study clearly guarantees the feasibility of our analog Kalman filter circuit. However, Ref. [21] did not provide any comparative analysis with/without analog-digital hybrid computing in the sensor fusion process.

- 2) Ref. [21] mainly focused on the improvement of "speed" (response time) in robot control, where PD controller operation was accelerated through analog-digital hybrid computing. In contrast, our study mainly focused on the improving the accuracy and energy-efficiency in sensor fusion through analog-digital hybrid computing. Therefore, the goals of Ref. [21] and our study are completely different.

We revised the CONCLUSIONS section to clarify the originality of our work more clearly.

Revised CONCLUSION: In this study, an analog–digital hybrid computing platform was demonstrated for **low-power and high-accuracy** sensor fusion in drones. The hybrid computing platform was built by the co-integration of the conventional digital processor (digital component) and SnS₂ memtransistor-based analog circuit (analog component). The developed continuous-time Kalman filter algorithm was implemented using an analog Kalman filter circuit, where the precise tunability and high reliability of the SnS₂ memtransistor enabled stable algorithm execution. **The output of the analog Kalman filter circuit shows good agreement with the result of a traditional software-based Kalman filter, which provides a practical accuracy of analog-digital hybrid computing.** In addition, our hybrid computing platform achieves higher energy efficiency while mitigating the burden on the battery capacity and computing power of the digital component. Notably, because the memtransistor does not require to be reprogrammed frequently and is non-volatile, the power consumption required to optimize the performance of the computing platform can be minimized. Consequently, the memtransistor-based analog–digital hybrid computing platform can be applied to a broad spectrum of applications that require processing sensing data with limited energy, such as the Internet of Things and edge computing applications.

We believe that the power consumption of our hybrid computing platform can be further reduced when the sensor module is integrated directly into the analog Kalman filter circuit. Because the measured sensing signal is analog, the analog Kalman filter circuit can process the data directly without using any analog-to-digital conversion, thereby minimizing both the latency and quantization error. **In addition, because the memtransistor has one more electrode (gate electrode) than a conventional two-terminal memristor, the device conductance can be adjusted through the gate electrode, which enables simpler circuit configuration. A simple circuit configuration is also expected to contribute in reducing the overall power consumption.**

Reviewer #2:

The authors reported a hybrid circuit of sensor fusion for the rotation motion of a drone. Most importantly, they demonstrated the Kalman filter using SnS₂ "memristor" based analogue circuits. They also showed this approach has lower latency and is more energy efficient. This is a very good work, however, there are still a few issues that need to be addressed.

Response: We appreciate the careful and constructive comments from the reviewer. We have provided point-by-point responses to the reviewer's comments below. Our responses are indicated in blue colored text, and the additions (or revisions) to the manuscript are marked in green colored text.

1) Memristor is a 2-terminal device. The SnS₂ device is actually a 3-terminal device, not really a memristor. Of course, they used the device like a 2-terminal memristor with a fixed gate voltage. To prevent confusion, it would be nice to clarify this.

Response: Thank you for your comment. As suggested, we have revised INTRODUCTION section to clarify the difference between a memristor and memtransistor.

Revised INTRODUCTION (page 5, last paragraph): In this study, we demonstrate a memtransistor (memristor with transistor structure²²)-based analog-digital hybrid computing platform for sensor fusion with higher energy efficiency. The measured data from both the gyroscope and accelerometer were combined to accurately determine the Euler angles of drones, wherein the Kalman filter algorithm was implemented using a customized analog circuit with the memtransistor. Because this analog Kalman filter circuit can operate independently without using the computing resources of the microcontroller, the computational burden on the microcontroller is reduced, and subsequently a reduction in overall power consumption can be expected. Here, we used transition-metal dichalcogenide (TMD) materials in the channel of the memtransistor, which is a three-terminal hybrid memristor and transistor. The bulk traps located at the tin disulfide (SnS₂) nanosheet exhibit a highly reliable nonvolatile resistive switching behavior, which is achievable through the electrical pulse applied to a gate electrode. Finally, we showed that a drone with hybrid computing performs sensor fusion with higher energy efficiency than a drone with only a conventional digital processor.

2) To implement the Kalman filter properly, the resistance of each "memristor" needs to be independent of the voltage on the device. Therefore, a linear I-V curve is preferred. Yes, they included the IV curves

at fix gate voltages. But the Kalman filter is worked with $V_g=0$, and the $V_g=0$ curve is too hard to see. A zoom-in view of $V_g=0$ curve is needed.

Response: Thank you for your comment. As shown in Fig. R6, a linear relationship exists between V_D and I_D at $V_G = 0$ V. We have added this graph as the inset of Fig S6a.

Figure R6. Measured I_D - V_D curve at $V_G = 0$ V.

3) If the authors could compare this SnS_2 device with other types of memristors and explain why this device is used, it would be useful.

Response: Thank you for your comment. Because the three-terminal memtransistor can control the device conductance by using the gate electrode, the peripheral circuit configuration for the memtransistor can be simplified more compared to that for the two-terminal memristor.

Figure R7. (a) Memristor with two electrodes. (b) Example analog circuit with the memristor. (c) Memtransistor with three electrodes. (d) Example analog circuit with the memtransistor.

The memristor contains only two electrodes, thereby V_{write} (i.e., voltages for adjusting the conductance) must be applied to both ends of the two electrodes (Fig. R7a). However, these two electrodes are also used for the signal input/output path. Consequently, additional peripheral circuits are required to distribute the input signal ($E_{mea}(t)$) and V_{write} properly (Fig. R7b).

In contrast, the memtransistor has three electrodes, and the conductance can be adjusted by using the gate electrode independently (Fig. R7c). Therefore, the conductance can be adjusted without additional peripheral circuits (Fig. R7d). In summary, there is no difference between the memristor and the memtransistor in the conductance change characteristic itself, but the memtransistor enables simpler circuit configuration. However, because this study did not verify this advantage experimentally, a brief discussion is added in the CONCLUSION as follows:

Revised CONCLUSION: We believe that the power consumption of our hybrid computing platform can be further reduced when the sensor module is integrated directly into the analog Kalman filter circuit. Because the measured sensing signal is analog, the analog Kalman filter circuit can process the data directly without using

any analog-to-digital conversion, thereby minimizing both the latency and quantization error. In addition, because the memtransistor has one more electrode (gate electrode) than a conventional two-terminal memristor, the device conductance can be simply adjusted through the gate electrode, which enables simpler circuit configuration. A simple circuit configuration is also expected to contribute to reducing the overall power consumption.

4) In the paper, the system is built on bread boards. It is a good concept demonstration; however, it is too big/heavy for drone. If they can put the system on a PCB, it would be nice.

Response: Thank you for your comment and we completely agree with it. In this work, for the proof-of-concept of our analog-digital computing platform, we implemented the analog circuit on a bread board.

In fact, to integrate our analog Kalman filter circuit on a PCB and mount it on the drone, 1) a circuit for generating a voltage pulse, 2) a circuit for performing the update-verify feedback process, and 3) a circuit for power distribution are additionally required. Therefore, further research on the design of these peripheral circuits is necessary. We hope that the system-level analog-digital hybrid computing platform will be demonstrated in a near-future study.

REVIEWERS' COMMENTS

Reviewer #1 (Remarks to the Author):

Thank you for addressing my comments. I am generally happy with your responses. Nevertheless, while the response is adequate, the change in the manuscript are not sufficiently reflecting these. Some comments while they have been addressed in the response letter, they have not been addressed in the main article. For example Fig. R1 and the related discussions would greatly enhance the paper, either in the main part of the paper or more suitably in the SI. The same goes for Fig, R5. A couple of sentences addressing comment 8 should also be added. A lot of the information discussed in your response to comment 12 should also be added in the main manuscript to strengthen the novelty, etc.

Reviewer #2 (Remarks to the Author):

The authors have addressed all concerns that I raised.

Reviewer #1:

Thank you for addressing my comments. I am generally happy with your responses. Nevertheless, while the response is adequate, the change in the manuscript are not sufficiently reflecting these. Some comments while they have been addressed in the response letter, they have not been addressed in the main article. For example Fig. R1 and the related discussions would greatly enhance the paper, either in the main part of the paper or more suitably in the SI. The same goes for Fig, R5. A couple of sentences addressing comment 8 should also be added. A lot of the information discussed in your response to comment 12 should also be added in the main manuscript to strengthen the novelty, etc.

Response: We appreciate the careful and constructive comments received from the reviewer. According to the reviewer#1's comments,

- 1) Fig. R1 and the related discussions are included as Supplementary Information Note 7.
- 2) Fig. R5 and the related discussions are included as Supplementary information Note 4-6
- 3) Comment #8 is added to the Supplementary information Note 5-2.
- 4) Comment #12 is added to the main manuscript in DISCUSSION section.

Reviewer #2:

The authors have addressed all concerns that I raised.

Response: We appreciate the positive comment from the reviewer.